# Cooperative growth in microbial communities is a driver of multistability

William Lopes [1] ✉, Daniel R. Amor [1,2,3,4] & Jeff Gore [1] ✉

Microbial communities often exhibit more than one possible stable composition for the same set of external conditions. In the human microbiome, these persistent changes in species composition and abundance are associated with health and disease states, but the drivers of these alternative stable states remain unclear. Here we experimentally demonstrate that a cross-kingdom community, composed of six species relevant to the respiratory tract, displays four alternative stable states each dominated by a different species. In pairwise coculture, we observe widespread bistability among species pairs, providing a natural origin for the multistability of the full community. In contrast with the common association between bistability and antagonism, experiments reveal many positive interactions within and between community members. We find that multiple species display cooperative growth, and modeling predicts that this could drive the observed multistability within the community as well as non-canonical pairwise outcomes. A biochemical screening reveals that glutamate either reduces or eliminates cooperativity in the growth of several species, and we confirm that such supplementation reduces the extent of bistability across pairs and reduces multistability in the full community. Our findings provide a mechanistic explanation of how cooperative growth rather than competitive interactions can underlie multistability in microbial communities.

From driving Earth's biogeochemical cycles[1] to maintaining human health[2,3], microbial communities play essential functions across ecosystems. Such functions are closely linked to the abundance dynamics of diverse community members, which typically include taxa from multiple kingdoms[4]. Microbial communities can exhibit multistability, the potential for microbial abundances to reach alternative stable states under the same environmental conditions[5]. For instance, the human microbiome exhibits signatures of multistability[6], such as a bimodal distribution of *Prevotella* abundances in the gut, with one mode corresponding to the bacteria dominating the community, while in the other mode it is present in a low-abundance state[7]. In turn, the presence of multistability can enable short-term, temporary

perturbations to induce sharp and lasting transitions in community composition and function[8–12]. Persistent shifts in microbiome composition can significantly impact human health, potentially leading to unhealthy states that may prove difficult to reverse even with antibiotic treatment[13]. Both of these characteristics—dramatic shifts in species composition combined with difficulty in reversing the shift—are considered hallmarks of multistability[14]. Despite the fundamental importance of such alternative stable states, the drivers of multistability in microbial communities remain relatively unknown.

Strong mutual antagonism is among the most recognized drivers of bistability across many systems, including microbial systems[15–18]. In pairwise competitions, inhibitory interactions among *Streptomyces*

[1]Physics of Living Systems, Department of Physics, Massachusetts Institute of Technology, Cambridge, MA, USA. [2]Institute of Biology, University of Graz, Graz, Austria. [3]Present address: LPENS, Département de physique, Ecole normale supérieure, Université PSL, Sorbonne Université, Université Paris Cité, CNRS, Paris, France. [4]Present address: IAME, Université de Paris Cité, Université Sorbonne Paris Nord, INSERM, Paris, France. ✉e-mail: lopesw@mit.edu; gore@mit.edu

strains help antibiotic-producing isolates resist invasions while at high abundance, thus contributing to bistability[19]. Production of inhibitory compounds is analogous to other combat strategies, such as the production of colicins by *Escherichia coli*, that prevent competing strains from invading a niche. An experimental system composed of two different strains *of E. coli* showed that a colicin-producing strain could survive and kill a fast-growing sensitive competitor, though only if the slow growing strain surpassed an abundance threshold[20]. This phenomenon was recapitulated in a simple theoretical model, and similar models have been successful in predicting bistable outcomes in different microbial communities[21,22]. The well-known, generalized Lotka-Volterra model can successfully explain the alternative outcomes of in vitro competitions between different sets of bacterial isolates from the human gut[23], although this simple model can also fail to capture complex microbial interactions[24,25]. While competitive interactions provide a canonical explanation for alternative stable states in microbial communities, their relative importance in comparison to other potential drivers remains largely unexplored.

Within-species interactions such as cooperative growth can also play an important role in shaping community structure[26–28]. Such dynamics arise from the need for a critical number of individuals in order to achieve efficient growth, leading to a maximum *per capita* growth rate at intermediate population density. Indeed, some cooperatively growing populations undergo extinction if population density falls below a critical value—termed the Allee threshold[29]. Populations with positive density dependence in their *per capita* growth may have a disadvantage when competing against populations with noncooperative growth, especially when starting at lower abundances[22,27]. Yet, it is still unclear to what extent the joint effects of species with different growth dynamics can shape multistability beyond pairwise interactions.

Here, we assembled a cross-kingdom community from six-species relevant to the human respiratory tract and found that it displayed a remarkably high degree of multistability—characterized by four alternative stable states. We set out to determine whether pairwise-based rules of community assembly[21] could capture the six-species community dynamics. We found widespread bistability among pairs that explained the alternative stable states of the full community. In contrast to the expectation that bistability is generated by strong mutual inhibition, spent media experiments revealed many cooperative interactions within and between community members, leading us to uncover cooperative growth dynamics among focal species. The tractability of our experimental system enabled the finding that glutamate supplementation weakens the cooperative growth of multiple species. Consistent with predictions of dynamical systems modeling of microbial interactions, we found that glutamate supplementation reduces the number of alternative stable states both in pairs and in the full community, thus demonstrating that cooperative intraspecies growth is a primary driver of multistability in the community.

## Results

### Multistability in a microbial community

To understand the assembly of a microbial community important to the human nasal microbiota, we selected a set of six cross-kingdom species that includes both commensals and pathogens found in the human respiratory tract as a model system (Fig. 1a): *Cryptococcus neoformans* (Cn), *Corynebacterium pseudodiphtheriticum* (Cp), *Lactiplantibacillus plantarum* (Lp), *Moraxella catarrhalis* (Mc), *Staphylococcus aureus* (Sa), and *Staphylococcus epidermidis* (Se). We started by inoculating replicate communities of all six species cocultured at equal initial abundances in modified 10% Brain Heart Infusion broth (BHI) and propagating them every 24 h for five growth–propagation cycles with a fixed dilution factor of 1:100 (Fig. 1b). After five growth cycles, we plated the communities onto a combination of selective and differential media to assay the community structure through CFU

(Colony Forming Units) counting. Surprisingly, we found that replicate communities assembled into widely divergent outcomes, with highly different relative abundances among species. Two replicate communities were dominated by *S. epidermidis* (Se), two communities were dominated by *S. aureus* (Sa), and two communities were composed in large part of both *S. epidermidis* (Se) and *C. neoformans* (Cn) (Fig. 1c). We therefore found that replicate communities with equal initial abundances of the six species display a surprisingly divergent set of final communities.

We hypothesized that the divergent outcomes might reflect the presence of alternative stable states in the community, given that the replicate communities were assembled under identical conditions and from the same inoculum of species. To test this hypothesis of multistability, we cocultured the six-species at different initial abundances, in which each one of the species was inoculated at an initial abundance of 95% and the remaining species at 1%. From these starting points, we found that the community assembly process became highly reproducible and revealed the presence of multistability within this community. In particular, initializing Cn, Mc, Sa, or Se at 95% of the community, we found that each of them could drive the five other species extinct, a result that was identical among the six community replicates (Fig. 1d). Furthermore, this observation suggests that communities starting at equal initial abundances might require more than 5 cycles to reach one of the four stable outcomes, as denoted by the presence of mutually exclusive species in the outcomes for communities 3 to 6 in Fig. 1c. Taken together, our findings uncover a surprisingly high degree of multistability in our six-species community, with four non-invadable and reproducible stable states.

### Pairwise competitions show bistability

Next, we asked whether pairwise interactions between community members could, through simple assembly rules[21], shape the observed multistability in the six-species community. We competed the focal species—Se, Sa, Mc, and Cn—that dominate the alternative stable states in all possible pairwise cocultures (Fig. 2a). We inoculated cocultures at two starting species abundances, in which one of the two species was initially dominant (99% of inoculated cells), and we measured species abundances every other day for five propagation cycles. To classify pairwise competition results, we assessed the three canonical equilibrium outcomes that occur within the competitive Lotka-Volterra model[30] (Fig. 2b, "Methods"): coexistence between two species at a stable fraction, exclusion of one species by the other independently of the initial abundance, or bistability in which the species at high-abundance excludes the low-abundance competitor. Remarkably, we found that all six pairs exhibited bistable relationships between competing species (Fig. 2c). In a majority of cases, extinction of the low-abundance competitor occurred within just one or two propagation cycles. The fully bistable network of pairwise interactions (Fig. 2d) recapitulated the same four stable states observed in the six-species community, suggesting that pairwise interactions can drive outcomes in the full six-species community.

### Focal species exhibit cooperative growth

To test whether mutually antagonistic interactions, canonical drivers of bistability, could underlie the observed competition outcomes, we performed a spent-medium assay in which species were inoculated in each other's spent media as well as their own (Fig. 3a). We found that the metabolic byproducts produced by Mc strongly inhibited all other species, in agreement with the hypothesis that an antagonistic interaction could allow Mc to overtake any of its three focal competitors. Conversely, we observed that the metabolic byproducts produced by Cn, Sa, and Se extensively facilitated the growth of competitor species in most cases (Fig. 3b). Taking these spent-media experiments as a prior for interspecies interactions in cocultures, the observed growth facilitation towards Mc argues against mutual antagonisms being the

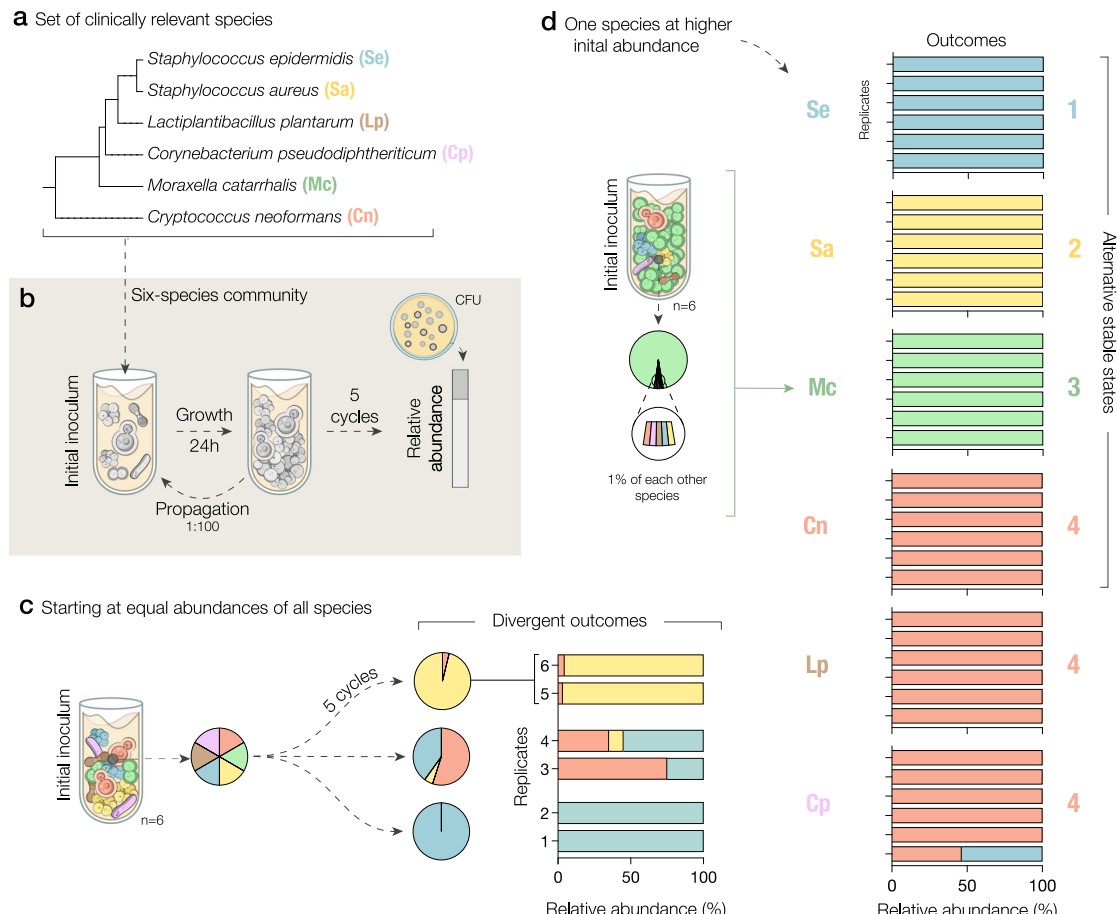

**Fig. 1 | A six-species community of clinically relevant species exhibits four alternative stable states. a** Selection of a cross-kingdom set of commensal and pathogenic species relevant to the human respiratory tract. The five bacterial species are displayed in a phylogenetic tree; followed by the yeast (bottom). **b** Experimental scheme: We competed all six-species starting at equal abundances for five daily propagation cycles. Final species abundances were determined by plating the communities onto selective and differential media. **c** Despite their equal starting abundances, these communities reached highly different compositions in distinct replicates. **d** Culturing communities with one species initially at high abundance (95% of cells) revealed four alternative stable states in which a different species (Se, Sa, Mc, or Cn) takes over the community. Bars show the final species abundances for each replicate of the six-species competitions, clustered by the initially dominant species (*n* = 6).

main driver of bistability in focal-species pairs. In addition, we tested if pH changes induced by bacterial growth, a common driver of microbial interactions[18,31], could generate unfavorable conditions leading to inhibitory interactions between species. We found that most species alkalinize the media independently of initial pH, making it unlikely that pH changes are driving inhibitory interactions (Supplementary Fig. 1). We concluded that antagonistic interactions are insufficient to explain bistability observed across focal-species pairs.

Seeking alternative drivers of the observed bistability, we next considered whether facilitation could shape the outcome of experimental pairwise competitions. In particular, the presence of self-facilitation, or cooperative growth, could interfere with the survival of a species at low density. To assess the presence and strength of cooperative growth across the focal species, we experimentally measured the *per capita* growth rates over a range of initial abundances in monoculture (Fig. 3c, Supplementary Fig. 2). We found that Cn is characterized by logistic growth (Supplementary Fig. 3), in which the *per capita* growth rate decreases monotonically with the population density. Sa and Se displayed a weak Allee effect, in which *per capita* growth rate is always positive but is reduced at low cell densities. Notably, distinct growth dynamics can affect the abundances of species over growth cycles in pairwise competitions. For example, in the Cn-Sa pair (Fig. 2c), the weak Allee effect in Sa results in slower initial growth at low abundance, allowing Cn to grow in

relative abundance before Sa reaches its maximum growth rate and carrying capacity. Finally, we found that Mc is subject to a strong Allee effect, which is characterized by a negative growth rate at initial densities below a threshold (~10^5 cell/mL). Interestingly, Mc's growth is strongly facilitated by all competitors even when cocultures are initialized below its survival threshold in monoculture (Supplementary Fig. 4). An analysis of the population density over propagation cycles as a function of the density on the first day enabled the identification of monoculture survival thresholds under daily dilutions (Supplementary Fig. 5). In summary, three out of four focal species exhibit cooperative growth, which could potentially shape the outcome of interspecies competition.

We used a simple theoretical model to gain insight into how the Allee effect could shape outcomes of pairwise competitions between focal species. Specifically, we considered a modified version of the generalized Lotka-Volterra model that accounts for an Allee effect acting on some of the species ("Methods"). For species subject to an Allee effect, the dynamics are given by:

$$\frac{dX_i}{dt} = r_i X_i \left( (X_i - a_i)(1 - X_i) - \alpha_{ij} X_j \right) \tag{1}$$

where $X_i$ stands for abundance of species *i* (normalized to its carrying capacity), $r_i$ stands for its *per capita* growth rate, $\alpha_{ij}$ captures the

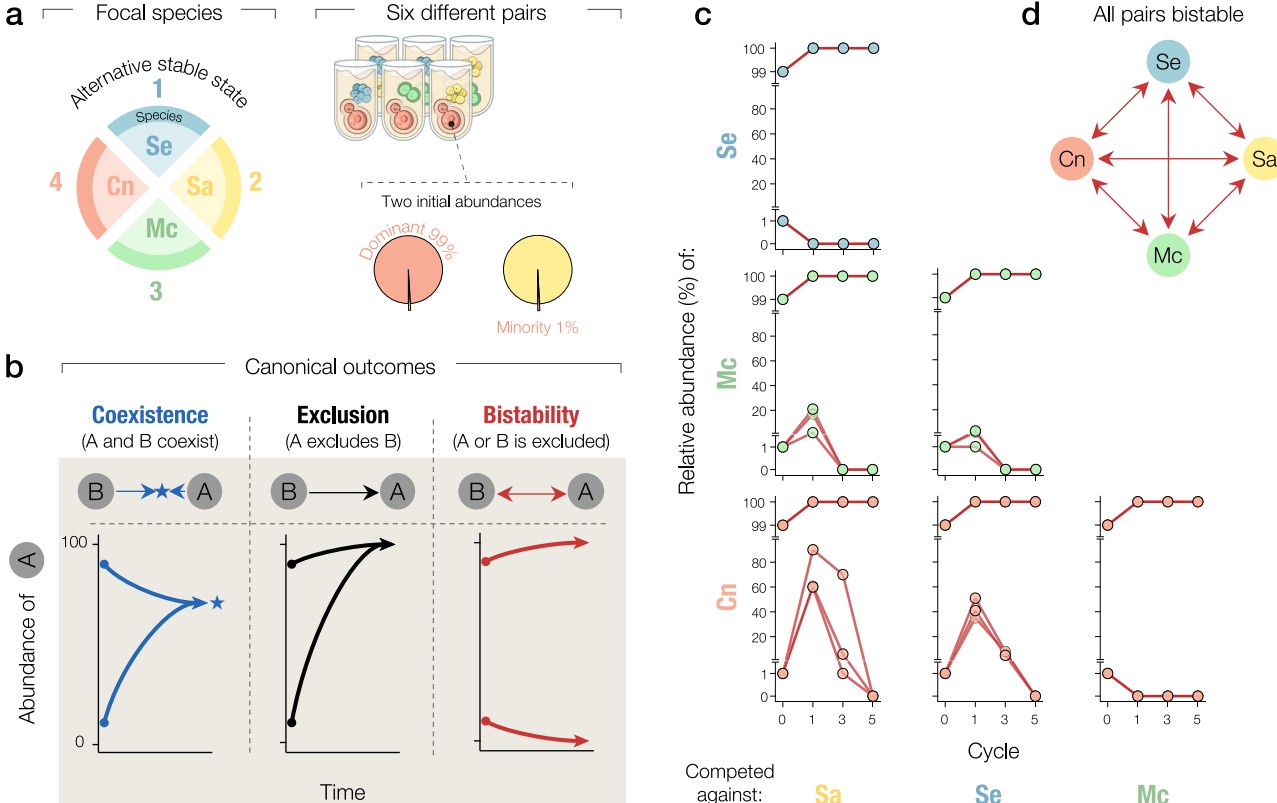

**Fig. 2 | Pairwise competitions reveal extensive bistability among species that dominate each one of the four alternative stable states. a** Experimental scheme: We performed pairwise competition experiments between the four focal species that dominate each of the alternative stable states. All species from the six different pairs were competed twice, as either the dominant (99% abundance) or the minority (1%) species. **b** Schematic cartoon of the three canonical outcomes of pairwise competition experiments illustrated through the dynamics of the relative abundance of one of the competitors. Coexistence (left): the abundance of both species converges to an intermediate value (star) and both species persist over time. Exclusion (center): one species takes over the other regardless of the initial abundance. Bistability (right): the winner of the competition is determined by the initial species abundance. **c** Time series of the relative abundance for all six pairs in competition experiments. **d** The interaction network of the focal species, as revealed by pairwise competitions experiments, shows extensive bistability: each member can win each of the pairwise competitions when initialized at high relative abundance. Each pairwise competition was replicated at least three times.

strength of the interaction from species $j$ towards species $i$ (with $i \neq j$) and $a_i$ captures the strength of the Allee effect. Note that a strong or weak Allee effect is captured by a positive or negative value of $a_i$, respectively. For those species free of the Allee effect we consider dynamics given by the classical version of the Lotka-Volterra model, i.e., they are subject to logistic growth. In this model, the strong Allee effect, under which a species such as Mc fails to survive at low densities, can allow competitors to take over relatively independently of competitive interspecies interactions. Notably, even the weak Allee effect, under which a species exhibits strictly positive growth rates in monoculture, can turn a scenario of competitive exclusion into bistability (Fig. 3d). In particular, the strength of the Allee effect can determine the position where the nullclines, boundaries between net positive and negative growth of a species in the phase space, cross. While a strong Allee effect can generate bistability in pairwise outcomes regardless of the sign of interspecies interactions, the weak Allee effect can only drive such bistability under competitive interspecies interactions (Supplementary Fig. 6). Therefore, even if a species exhibits only positive net growth at low abundance in monoculture, as does Se (Fig. 3c), its cooperative growth can potentially lead to the emergence of bistability in coculture (Fig. 3d).

### Cooperative growth drives multistability

To test whether cooperative growth dynamics indeed drive the widespread bistability in competition experiments, we considered whether the strength of the Allee effect could be tuned by nutritional supplementation. We began by performing a biochemical screening assay,

which consisted of supplementing BHI with 10 mM of 19 molecules important to central metabolic pathways including glycolysis and the tricarboxylic acid (TCA) cycle. We measured the fold growth ($OD_{24h}/OD_{0h}$) of the focal species at low inoculum densities and found that glutamate supplementation markedly increased the fold growth of all species (Fig. 4a, Supplementary Fig. 7). Next, we tested the effect of glutamate on species' *per capita* growth rates over a range of initial abundances and found that glutamate supplementation eliminated the weak Allee effect in both Sa and Se and lowered the survival threshold in Mc (Fig. 4b).

After validating glutamate as an experimental lever to modulate cooperative growth of three focal community members, we undertook a theoretical analysis of the potential impact of reducing cooperative growth in pairwise competitions. We found that reducing the strength of the Allee effect could reshape competition outcomes in different ways (Fig. 5a, b). In cases in which one competitor grows logistically (case of Cn, "Methods"), reducing the Allee effect acting on the other competitor could lead to a competitive exclusion scenario in which the second competitor dominates. In cases in which a species is subject to the strong Allee effect (Mc) while its competitor exhibits a weak Allee effect, reducing the weak Allee effect can lead to bistability in which one of the stable states enables coexistence of the competitors. Surprisingly, tristability can emerge between two species that are subject to both weak interspecies interactions and a weak Allee effect. In this case, a third stable state emerges in which competitors coexist, while the two canonical states of bistability remain as available equilibria. Weakening the Allee effect for both species can easily transform such

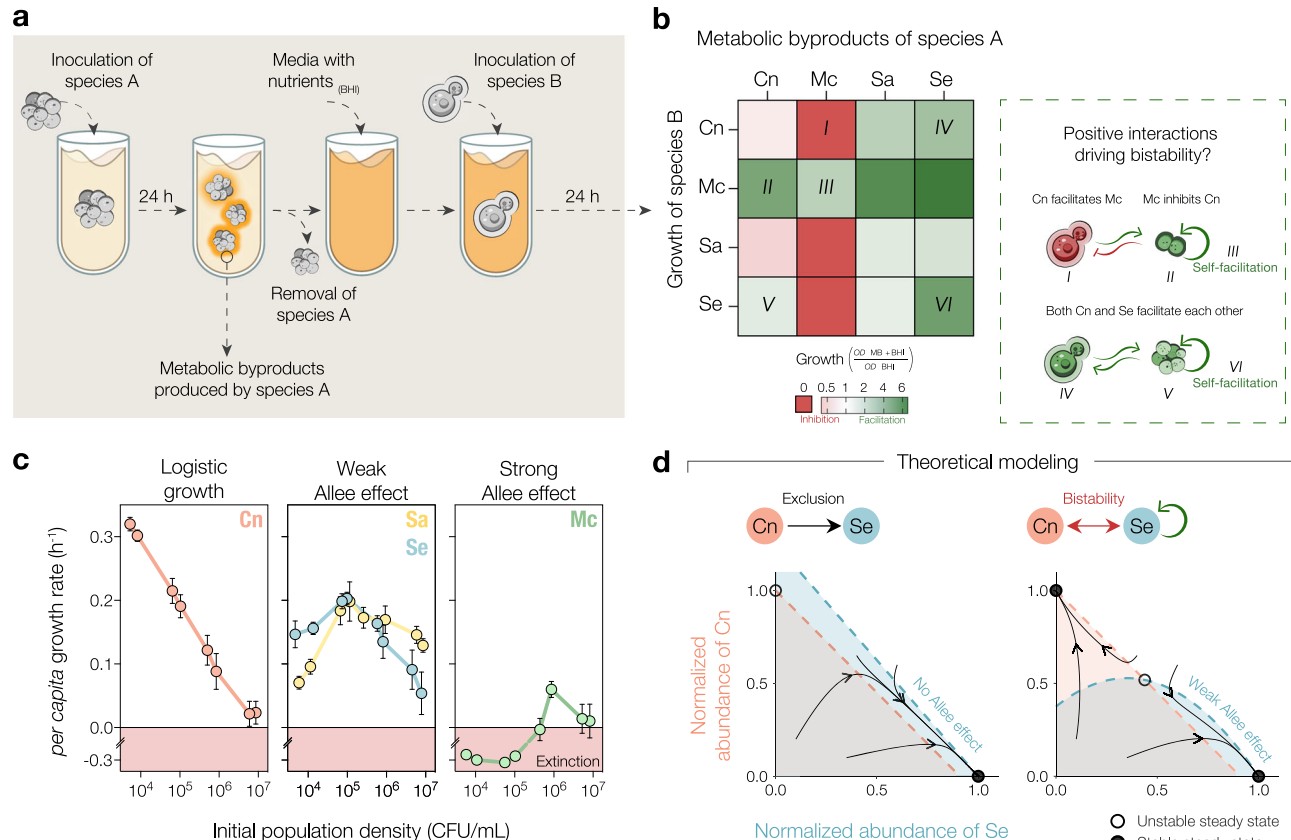

**Fig. 3 | Most of the focal species exhibit intra- and inter-species facilitation, which could theoretically drive bistability through the Allee effect.**
**a** Experimental scheme: Focal species were grown as monocultures for 24 h, and cells were filtered out from the culture. All focal species were grown in other species' spent media, as well as their own (see "Methods"). **b** The heat map of species' growth in spent media shows both inhibitory and facilitative interspecies interactions, as well as cooperative growth or self-facilitation, with examples highlighted in the box on the right. Values represent the average of at least three measurements. **c** Growth rates for the focal species as a function of the initial population density (see "Methods") reveal a logistic growth pattern in Cn, whereas a positive but compromised growth rate at low densities indicates that both Sa and

Se are subject to the weak Allee effect. Mc displays the strong Allee effect leading to population extinction below a survival threshold of $10^5$ cell/mL. Error bars are the standard error (SE) across replicates, $n = 3$. **d** Exclusion and bistability outcomes as predicted by a Lotka-Volterra model in which either Se grows logistically (left) or exhibits a weak Allee effect (right). We model the growth of Cn as logistic in both cases. Dashed lines show the nullclines, boundaries between net positive and negative growth rates, for Se and Cn as a function of their normalized abundances. The area under each nullcline, matched in color, indicates the region of positive *per capita* growth rate for the corresponding species. Arrows show the time trajectories of pairwise competitions starting from different initial abundances, empty dots show unstable steady states, and full dots show stable steady states.

tristability into a canonical coexistence outcome. Importantly, these two interspecies outcomes (tristability and bistability with a coexisting state) are not present within the classical Lotka-Volterra interspecies competition model[30]. Since competition outcomes depend on interspecies interactions in addition to the strength of the Allee effect, there is a range of cases in which qualitative outcomes are unchanged upon reducing the Allee effect (Fig. 5b). We performed additional experiments under glutamate supplementation (Fig. 5c) to test for potential changes in pairwise competition outcomes between focal community members. Remarkably, by reducing the strength of the Allee effects in Sa, Se, and Mc, we experimentally validated a wide range of theoretically predicted changes in pairwise competition outcomes including the reduction in number of bistable pairs (Fig. 5d). This also included the breakdown of tristability, a relatively complex outcome we had identified and characterized in additional cocultures of the Sa-Se pair (Supplementary Fig. 9). The presence of the Allee effects can therefore lead to a wide range of additional stable states in two-species competitions, including the non-canonical outcomes of tristability and bistability with one coexisting state.

Finally, we explored whether the reduction of bistability in pairs can propagate up to reduce the degree of multistability in our six-species community. We again initialized the community at six different initial abundances, each with a distinct species at 95% initial

abundance. After five growth dilution cycles in media with glutamate supplementation, we found that the community displayed only two stable states (Fig. 5e), as compared to the four stable states found in the absence of glutamate supplementation. Interestingly, these two stable states are consistent with the predictions from the assembly rules based on the pairwise outcomes. One of the two stable states consists of the coexistence of Sa and Se, whereas the other state consists of the coexistence of Sa and Mc. These results demonstrate that both weak and strong Allee effects can drive alternative stable states in complex microbial communities.

## Discussion

In this study, through the combination of experiments and theoretical modeling, we demonstrated that cooperative growth can underlie multistability in a cross-kingdom community composed of pathogenic and commensal microbial species. While recent studies have uncovered mechanisms of competition, such as antagonistic interactions[18] and resource competition[15], underlying alternative stable states in microbial communities, our results suggest that cooperative interactions can also be a common driver of multistability in microbial communities. In our microcosm experiments, we observed that cooperative growth dynamics in three focal community members led to the emergence of two out of four alternative stable states in a six-

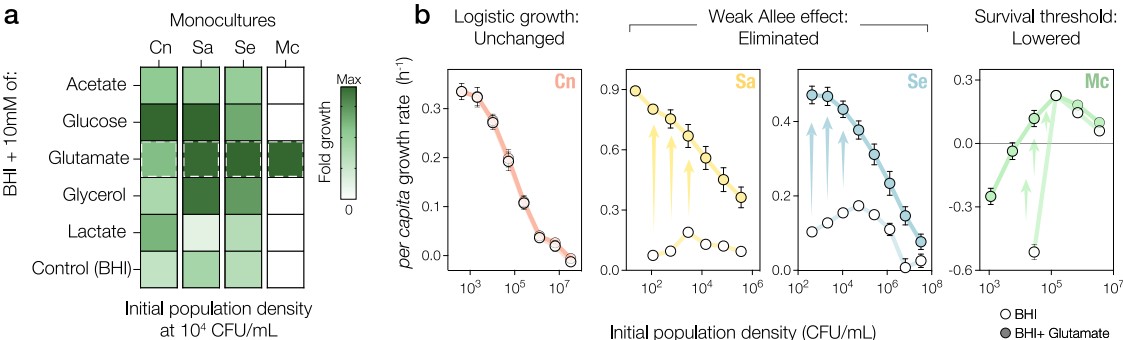

**Fig. 4 | Biochemical screening reveals that glutamate supplementation reduces the strength of cooperative growth. a** Subset of molecules used to supplement BHI in which focal species were inoculated at low population density ($10^4$ CFU/mL). Heatmap shows the fold growth ($OD_{24h}/OD_{0h}$, $n = 3$) in supplemented BHI and control (BHI only, bottom); glutamate (dashed borders) markedly increases the fold growth of all species with cooperative growth dynamics, allowing for Mc's growth even below its survival threshold ($10^5$ CFU/mL). Maximum fold growth is scaled for each column relative to monoculture of the indicated species. **b** Effect of glutamate supplementation on the growth dynamics of each focal species at different initial population densities. Panels show *per capita* growth rates as a function of the initial population density in BHI (empty dots) and BHI supplemented with glutamate (filled dots) at 100 mM, which is the dose that produced maximum fold-growth in a dose-response assay (Supplementary Fig. 8). Cn maintains its logistic growth pattern irrespective of glutamate supplementation, whereas Sa and Se adopt an analogous, monotonically decreasing growth pattern upon elimination of the weak Allee effect. Glutamate supplementation also lowers the survival threshold of Mc. Error bars are the standard error (SE) across replicates, $n = 3$.

species community, in addition to a wide range of non-canonical outcomes—including tristability—in pairwise competitions.

We showed that competing a six-species community at equal initial abundances leads to substantial outcome variability among replicates. The surprising variability could arise from stochastic fluctuations in initial population abundances (Supplementary Figs. 10 and 11), indicating that the community could exhibit extreme sensitivity to the initial species abundance[32], as observed in other experimental communities[15,33]. To provide an understanding of how species abundances affect community structure, we competed the six-species community with various initial abundances. We found that the microbial dynamics became highly deterministic, showing four repeatable patterns of community structure, in which the initially dominant species excluded all competitors—quadstability. This multistability could be explained by the widespread bistability of pairwise competitions, in agreement with simple rules of community assembly[21,34]. Coarse-grained rules have also proven effective in predicting outcomes such as collapse, coexistence and productivity, across models and experiments for mutualistic communities[35]. Guided by coculture outcomes (Fig. 2c), we used competitive interspecies interactions to theoretically analyze the impact of cooperative growth in bistable pairs (with an extension to mutualistic pairs provided in Supplementary Fig. 6). Future work could focus on addressing the impact of cooperative growth in experimental, mutualistic communities.

While examples of the Allee effect have been found in many species across a wide range of taxa, little attention has been given to the role of cooperative growth in clinically relevant communities. We found that three core bacterial members of the nasal microbiome display the Allee effect. Generally, many microorganisms exhibit advantageous traits that require high population densities[36–38]. In pathobiont species such as Sa, a minimum population density is often required to initiate expression of virulence factors needed to establish successful infections[39]. The alternative stable states observed in this study consistently exhibit low biodiversity, constraining comparative analysis with typical states of the human microbiota. However, the Allee effect may nonetheless play a crucial role in driving alternative stable states in complex communities.

Our work established a tractable experimental system where the impact of cooperative growth in shaping community structure can be tested. We found that the strength of the Allee effect can be controlled by tuning the concentrations of glutamate in the growth medium. This is consistent with the demonstration that glutamate supplementation can eliminate growth defects and promote the growth rate of bacterial species, including Mc, under poor nutritional conditions[40]. In Sa, it has been demonstrated that glutamate—and amino acids that can be converted into glutamate—serves as a central carbon source that promotes growth in media lacking a preferred carbon source[41]. In addition, inhibition of glutamate biosynthesis was shown to disrupt biofilm formation in bacteria[42]. Given that the concentration of glutamate in nasal secretions is high in comparison to other amino acids[43,44], it is plausible to assume that glutamate may influence the growth dynamics and long-term survival of some microbes colonizing the nasopharynx. Finally, our system demonstrated that reducing cooperative growth also reduces the extent of bistability in pairwise competitions and the number of alternative stable states in the six-species community.

The community considered here provides a simplified representation of communities that play a critical role in human health. Of notable clinical relevance, ~30% of healthy people asymptomatically carry a strain of Sa in the anterior nares[45], a significant observation, since the vast majority of cases of Sa bloodstream infection appear to be endogenous, involving the same strain as the nasal colonizer[46,47]. Moreover, several studies have shown that during infections such as pneumonia, there is a reduced diversity in the respiratory microbiome often marked by increased abundance of Mc. Following an infection, individuals may establish a long-term, healthy microbiome with a composition shift towards the dominance of Mc[48,49]. Our simplified model community provides a mechanistic explanation of how cooperative growth can drive multistability in multispecies contexts. Further work will be necessary to determine the prevalence of multistability in the human respiratory microbiome and the extent to which cooperative growth drives it.

## Methods
### Reference strains
The six species chosen for this work were: *Cryptococcus neoformans* H99 (Cn, ATCC 208821), *Corynebacterium pseudodiphtheriticum* (Cp, ATCC 10700), *Lactiplantibacillus plantarum* (Lp, ATCC 8014), *Moraxella catarrhalis* (Mc, ATCC 25240), *Staphylococcus aureus* (Sa, ATCC 29213), and *Staphylococcus epidermidis* (Se, ATCC 14990).

### Media and inocula
All chemicals were purchased from MilliporeSigma unless otherwise stated. Pre-cultures were performed in Brain Heart Infusion Broth (BHI,

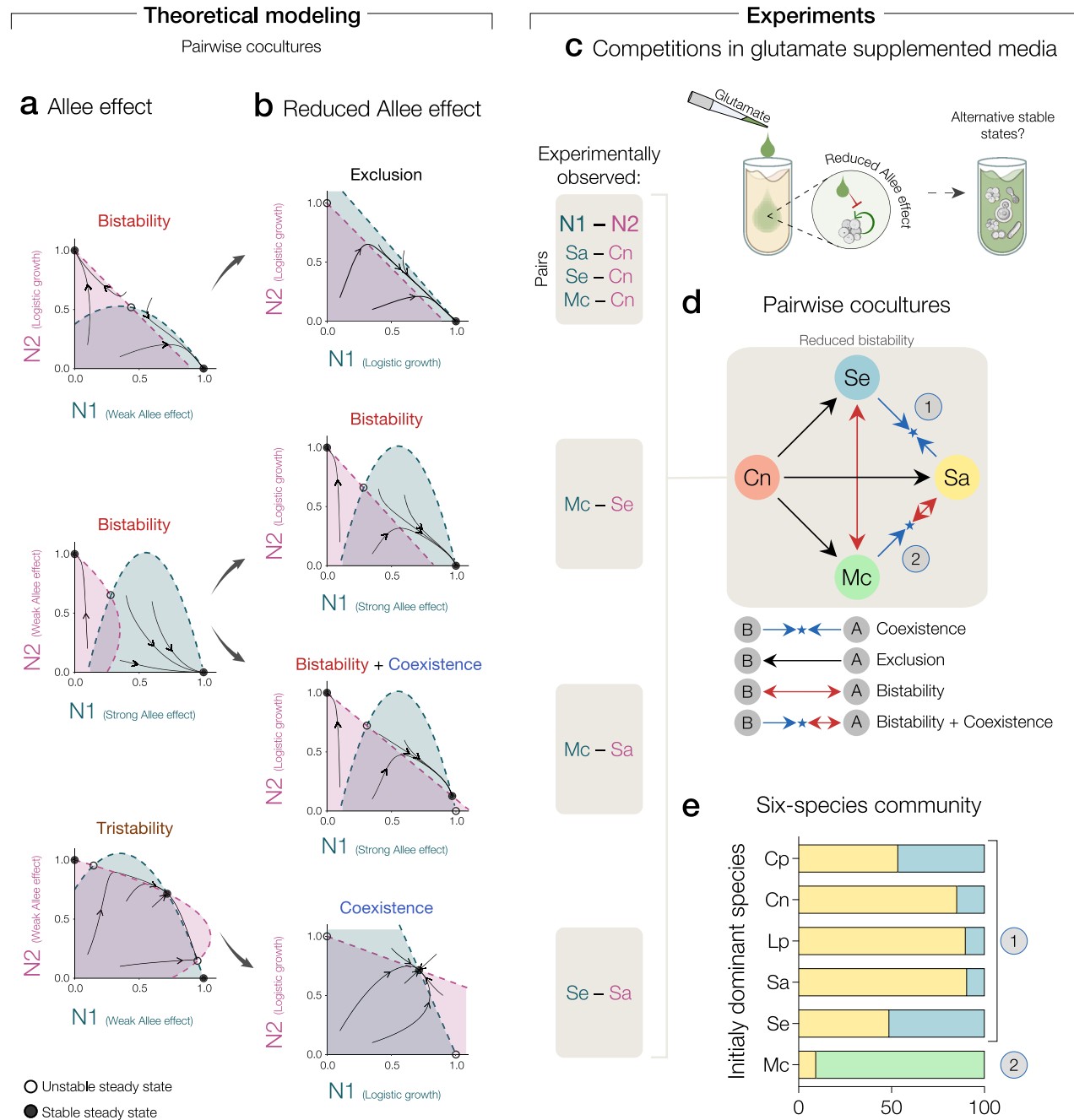

**Fig. 5 | Theoretical prediction and experimental demonstration of a decrease in the number of alternative stable states when the strength of cooperative growth is reduced.** Predicted and observed qualitative outcomes of pairwise competition, reshaped by reducing the strength of the Allee effect. **a** The left panels show theoretical phase planes under the Allee effect (without glutamate supplementation), while right panels **b** show the same under reduced Allee effect (with glutamate supplementation). Nullclines, trajectories, and steady states depicted by dashed lines, arrows, and dots, respectively. The area under each nullcline, matched in color, indicates the region of positive *per capita* growth rate for the

corresponding species, as shown in Fig. 3c. **c** To understand how the modulation of cooperative growth impacts the alternative stable states, we cocultured all pairs and the six-species community in media supplemented with glutamate. **d** We found a strong reduction in the number of bistable pairs, as compared to Fig. 2d. The shaded box on the left indicates the species pairs for which we observed the predicted reshaping of competition outcomes. **e** Under reduced Allee effect by glutamate supplementation, the six-species community displays two alternative stable states in which two species coexist (gray circles 1 and 2), as predicted from simple assembly rules based on pairwise competition outcomes[21].

Millipore 53286) containing: 12.5 g/L calf brains, 5.0 g/L beef heart, 10 g/L peptone, 5.0 g/L NaCl, 2.0 g/L D(+)-Glucose, and 2.5 g/L $Na_2HPO_4$. Competition experiments were performed in 10% BHI base medium supplemented to a final concentration of 10 mM phosphate (added as $Na_2HPO_4$/$NaH_2PO_4$), 5 mM urea, and 6.6 g/L NaCl. These supplementations were chosen to mimic the nasal environment regardless of phosphate concentration, buffering capacity, and urea

concentration[43]. All media were filter-sterilized using Bottle Top Filtration Units (MilliporeSigma).

**Competition experiments**

Frozen stocks of individual species were streaked out on BHI agar Petri dishes, grown at 37 °C for 48 h, and stored at 4 °C for up to 1 week. Before competition experiments, single colonies were picked and each

species was grown separately in 10 mL of BHI broth for 24 h. Cocultures were grown in 10% BHI broth, pH 7 to an $OD_{600}$ of $5 \times 10^6$ cell/mL, unless otherwise stated. $OD_{600}$ versus CFU experiments were performed previously to estimate adjustments for bacterial strains, while cells were counted using a Neubauer chamber for the yeast Cn. In pairwise competitions, diluted cultures were mixed to the desired starting abundances of 99%–1% and 1%–99% (Species A–Species B). In six-species competitions, experiments were initialized at equal species abundances or with one species prevalent at 95%. For competition experiments, cultures were grown in 96-deepwell plates (Deepwell plate 96/2000 μL; Eppendorf), with each well containing 600 μL of culture. Plates were covered with AeraSeal adhesive sealing films (Excel Scientific), incubated at 30 °C, and shaken at 1050 r.p.m. Every 24 h (propagation cycle), cocultures were mixed and diluted by a factor of 100 into fresh media with a 96-channel pipette (VIAFLO 96; INTEGRA Biosciences) using the program pipet/mix. $OD_{600}$ was measured at the end of each cycle, and final species fractions were estimated after five cycles of propagation. To measure final species fractions, cocultures were diluted and plated on selective/differential agar, as outlined below.

## Estimation of population density

For CFU counting, we serially diluted cultures in 0.9% NaCl by 10-fold dilutions (20 μL into 180 μL; maximum dilution: $10^{-7}$) with a handheld 96-channel pipette (VIAFLO 96; INTEGRA Biosciences) using the pipet/mix mode (mixing cycles: 5; mixing and pipetting speed: 8). From each well, 10 μL were pipetted onto the surface of 150 mm diameter agar plates using the reverse pipet mode (aspirate volume: 20 μL; dispense volume: 10 μL; dispense speed: 2). In two-species experiments, we used Standard Nutrient Agar 1 with Amphotericin B (15 g/L bacteriological peptone, 3 g/L yeast extract, 6 g/L NaCl, 1 g/L glucose, 4 mg/L Amphotericin B, and 18 g/L agar (STNA1-AmB), pH 7.2) for selective growth of bacteria, and Yeast Extract Peptone Dextrose Agar with chloramphenicol (20 g/L bacteriological peptone, 10 g/L yeast extract, 18 g/L agar and 0.05 g/L chloramphenicol (YPD-CHL), pH 6.7) for selective growth of the yeast Cn. In six-species experiments, we used a combination of modified compositions of selective or differential media for the species: Cn, Cp, Lp, Mc, Sa, and Se, allowing measurements of cell densities even at low abundances. We used a combination of:

- Trypticase Soy agar (15 g/L casein peptone, 5 g/L soya peptone, 5 g/L NaCl, 18 g/L agar, pH 7.2), supplemented with 65 g/L NaCl for the quantification of Sa and Se.
- STNA1-AmB agar as described above, supplemented with 5 mg/L vancomycin for the quantification of Mc.
- YPD-CHL agar as described above, for quantification of Cn.
- Man, Rogosa, and Sharpe agar (10 g/L bacteriological peptone, 5 g/L meat extract, 5 g/L yeast extract, 20 g/L glucose, 2 g/L $K_2HPO_4$, 2 g/L diammonium hydrogen citrate, 5 g/L sodium acetate, 0.1 g/L $MgSO_4$, 0.05 g/L $MnSO_4$, 1 g/L polysorbate 80, pH 5.5) supplemented with 4 mg/L AmB and 4 mg/L ciprofloxacin for the quantification of Lp.
- BHI-Tellurite agar (BHI 20 g/L, 0.5 g/L potassium tellurite, 44 mg/L L-Cystine, 18 g/L agar, pH 7.2) with 4 mg/L AmB for the quantification of Cp.

After plating, the 10 μL droplets were allowed to dry, and the plates were incubated at either 37 °C or 28 °C for bacteria/yeast growth. We performed CFU counting after 48 h of growth. Three or more plating replicates per condition were used.

## Cooperative growth dynamics

The species Cn, Mc, Sa, and Se were grown in 10 mL of BHI for 24 h at 30 °C. Cultures were centrifuged (10 min; 3220 × g; Eppendorf Centrifuge 5810R), washed in 0.9% NaCl and re-suspended to $10^7$ cell/mL in

modified 10% BHI, pH 7. The starting cell concentration was serially diluted in 96-deepwell plates (Deepwell plate 96/2000 μL; Eppendorf), covered with AeraSeal adhesive sealing films (Excel Scientific) and shaken at 1050 r.p.m at 30 °C. At 0, 6, 12, and 24 h, cultures were serially diluted as described above and plated on BHI agar to estimate CFU. Triplicates of CFU were counted after 48 h of incubation and final growth rate measurements were obtained from the slope of a regression line across all replicates.

## Spent-medium experiments

Species were grown in 10 mL of 10% BHI pH 7 for 24 h at 30 °C. Cultures were centrifuged (10 min; 3220 × g; Eppendorf Centrifuge 5810R) and spent media were sterilized with a 50 mL Steriflip Filtration Unit (0.22 μm; MilliporeSigma). 50 μL of spent media were spotted onto BHI plates to verify sterility. Spent media were used in place of water in modified 10% BHI, with all other reagents added back. A second preculture of each species was centrifuged (10 min; 3220 × g; Eppendorf Centrifuge 5810R) and re-suspended in each of four spent media at a starting concentration of $10^6$ cell/mL. 600 μL of cultures were then grown in 96-deepwell plates (Deepwell plate 96/2000 μL; Eppendorf) and shaken at 1050 r.p.m. for 24 h at 30 °C. Following 24 h of incubation, $OD_{600}$ in the different spent media were measured (Tecan Infinite 200 Pro plate reader) and divided by the $OD_{600}$ obtained in fresh media.

## pH measurements

The pH of microbial cultures was measured by using a pH microelectrode (Orion PerpHecT ROSS; Thermo Fisher Scientific).

## Theoretical model and simulations

To model the dynamics of the experimental communities, we used a modified version of the generalized Lotka-Volterra (gLV) model that accounts for the Allee effect acting on some of the species. In scenarios in which a given species $i$ was free of any Allee effect, the species dynamics corresponds to the classical gLV dynamics:

$$\frac{dX_i}{dt} = r_i X_i \left(1 - \alpha_{ij} X_j\right) \tag{2}$$

with $\alpha_{ii} = 1$. Note that, under this modeling choice, we consider that species grow logistically in the absence of an Allee effect.

In contrast, under the Allee effect the dynamics for a given species $i$ becomes:

$$\frac{dX_i}{dt} = r_i X_i \left((X_i - a_i)(1 - X_i) - \alpha_{ij} X_j\right) \tag{3}$$

where $i \neq j$, and $a_i$ captures the strength of the Allee effect.

Numerical simulations were performed in R, using the ordinary differential equations solver *ode* in the *DeSolve* package (vers. 1.40) and, for the case of stochastic simulations (Supplementary Figs. 10 and 11), the optimized tau-leap method in the *GillespieSSa* package (vers. 0.6.2).

Parameter values

To generate the panels in Fig. 3d we used the following parameter values:

$$\boldsymbol{r} = (r_1, r_2) = (0.2, 0.3) \, \text{hr}^{-1},$$

$$\boldsymbol{\alpha} = \begin{pmatrix} \alpha_{11} & \alpha_{12} \\ \alpha_{21} & \alpha_{22} \end{pmatrix} = \begin{pmatrix} 1 & 0.8 \\ 1.1 & 1 \end{pmatrix},$$

and a strength of the (weak) Allee effect $c_1 = -0.3$, with $i = 1$ and $i = 2$ corresponding to the species Se and Cn, respectively.

To generate the panels in Fig. 5a we used the parameter values stated below.

For the top panel:

$$\boldsymbol{r} = (r_1, r_2) = (0.2, 0.3)\, \mathrm{hr}^{-1},$$

$$\boldsymbol{\alpha} = \begin{pmatrix} \alpha_{11} & \alpha_{12} \\ \alpha_{21} & \alpha_{22} \end{pmatrix} = \begin{pmatrix} 1 & 0.8 \\ 1.1 & 1 \end{pmatrix},$$

$$c_1 = -0.3.$$

For the center panel:

$$\boldsymbol{r} = (r_1, r_2) = (0.2, 0.3)\, \mathrm{hr}^{-1},$$

$$\boldsymbol{\alpha} = \begin{pmatrix} \alpha_{11} & \alpha_{12} \\ \alpha_{21} & \alpha_{22} \end{pmatrix} = \begin{pmatrix} 1 & 0.2 \\ 1.2 & 1 \end{pmatrix},$$

$$\boldsymbol{c} = (c_1, c_2) = (0.2, -0.3).$$

For the bottom panel:

$$\boldsymbol{r} = (r_1, r_2) = (0.3, 0.3)\, \mathrm{hr}^{-1},$$

$$\boldsymbol{\alpha} = \begin{pmatrix} \alpha_{11} & \alpha_{12} \\ \alpha_{21} & \alpha_{22} \end{pmatrix} = \begin{pmatrix} 1 & 0.4 \\ 0.4 & 1 \end{pmatrix},$$

$$\boldsymbol{c} = (c_1, c_2) = (-0.3, -0.3).$$

To generate the panels in Fig. 5b we used the same parameters values as in Fig. 5a, while eliminating any weak Allee effect so that the corresponding species is subject to the classical gLV dynamics as described above. To model the reduction of the strong Allee effect in Mc in the center panels of Fig. 5b, we decreased its magnitude from $c_1 = 0.2$ to $c_1 = 0.1$. The outcome of bistability with coexistence in Fig. 5b was generated through changing the value of the interaction strength $\alpha_{21}$ from 1.2 into 0.9 (this change does not affect the qualitative outcomes in the center panel of Fig. 5a).

### Reporting summary

Further information on research design is available in the Nature Portfolio Reporting Summary linked to this article.

## Data availability

Source data are provided with this paper.

## Code availability

The code to generate the phase diagrams and the stochastic simulations is available at https://doi.org/10.5281/zenodo.10975215.

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

## Acknowledgements

J.G. acknowledges support from the Alfred P. Sloan Foundation and Schmidt Polymath Award. D.R.A. acknowledges the support of the Field of Excellence: Complexity in Life, Basic Research and Innovation at University of Graz. We thank all Gore Lab members for inspiring discussions.

## Author contributions

W.L., D.R.A., and J.G. conceptualized the project and designed experiments. W.L. carried out experiments and D.A. performed theoretical modeling. All authors analyzed the data and participated in writing and editing the manuscript.

## Competing interests

The authors declare no competing interests.
