## [Peer Review File · Nature Communications]

Cooperative growth in microbial communities is a driver of multistabilityREVIEWER COMMENTS

Reviewer #1 (Remarks to the Author):

This review is written by Jamila Roland-Chandler under the supervision of Wenying Shou.

Lopes et al. investigated the drivers behind multistability in the nasal microbiome using experiments and modelling. The authors identified 6-species microbial communities had multiple single species steady states, which were driven by bistability in two-species coculture. Self-facilitation, aka Allee effects, rather than mutual antagonism, drove bistability. Extending the Lotka-Volterra model to include Allee effects captured the observed dynamics in two-species coculture and predicted that weakening Allee effects would cause some loss of multistability and increased coexistence, which was validated experimentally. Therefore, Allee effects are important drivers of complex microbial community dynamics.

Lopes et al. explained a complex phenomenon in microbial community dynamics through experiments and modelling, and identified a novel mechanism for the emergence of multistability. Despite the complexity of the subject, the paper told a clear story that was easy to follow. Therefore, this paper should be accepted with minor corrections, which are detailed below.

Corrections

The Lotka-Volterra model with Allee effects should be included in the main text.

The introduction overpromises what multistable dynamics are captured in this paper. In the introduction, Lopes et al. describe how human microbiomes can exhibit multistability. For example, gut microbiomes can switch from a high diversity to low diversity state after application of antibiotics (line 36, reference 8). However, this paper does not explain that sort of multistability, as the alternative stable states described in this paper are single or

two species. A more appropriate example of multistability in the introduction should be used (e.g. reference 9 discussed bistability in the gut microbiome), or the discussion should include how the dynamics capture by this work could drive high and low diversity alternative steady states.

Using the term “cooperation” to describe intra-species cooperation/self-facilitation/Allee effects is confusing. Cooperation in the literature is used to describe intra- or inter-species cooperation, but Lopes et al. mainly use it to describe intra-species cooperation. Although they do specify this, I think this terminology is confusing and should be changed to self-facilitation.

Figure corrections

Fig. 1c and 5f: It is not clear which communities have reached steady state composition and which haven't. I think there should be plots of community composition over each cycle, like fig. 2c, to make this clearer.

Fig. 2c: In the Cn-Sa coculture, Cn increases from 1% to 80%, then declines to extinction. Why does this occur? I think this should be commented on.

Fig. 2d, 3d, 5d: The arrows used to indicate what steady states the system tends to are confusing. One could mistakenly think the arrows describe species interactions (like in fig. 3b). This representation should be changed. Also, for figures such as 3d, adding colour-matched + (growth) or – (death) above and below the nullcline will help readers to grasp the message quickly.

Fig. 5b and e: The shaded box indicating which pairwise cocultures exhibit theoretically-predicted dynamics is easy to miss. These results should be made more prominent. In addition, this information appears to be repeated in Fig 5e. Redundant information should be removed from the figure (e.g. add coculture names to fig. 5e like 5g).

Reviewer #2 (Remarks to the Author):

Analysis of the stable states is a central question in the study of microbial communities. This article looks into the multistability of synthetic microbial communities assembled from six species found in the respiratory tract. Bistability is commonly assumed to be driven by a mutually inhibitory interaction between two species (-/- interaction). This study, however, reports the presence of bistability in cooperating species (+/+) and in predator-prey type interaction (-/+), which is an interesting observation. The authors argue that cooperative growth can be a driving mechanism of bistability. They test and report the presence of cooperative growth among these species. Next, they use glutamate supplementation to tune down the level of cooperative growth which results in a reduction in the number of alternative stable states.

This article is generally well-written, clear, and easy-to-follow. Evidence is offered with well-designed experiments. We have the following minor comments that we think will be helpful for readers.

Minor comments:

1. The interaction values, α_{ij} , are all set to be positive which represent two competing species. However, the CFSM experiments in Fig 3b shows the presence of facilitation in many pairs. We would suggest additionally running the theoretical experiments under non-competing settings. In other words, setting the α_{ij} value in a way that its sign (- or +) matches Fig 3b.
2. The authors have observed three cases of strong/weak Allee effects and one case of logistic growth among the four main species that primarily determined the stable states. We think it would be helpful if they can speculate whether these proportion are commonly

expected. Under what conditions?

3. In their survey the authors found an important role for glutamate. Is this a general expectation? Is the concentration of glutamate particularly low or high in the respiratory tract? Alternatively, is BHI selectively low in glutamate? It would be nice to link back the observation to either the ecology of the respiratory tract or the particular laboratory conditions used for these investigations.

4. We would suggest including the raw data of the monocultures and cocultures used for the figures in the manuscript (either visually as supplementary figures or as a link to an openly available repository such as GitHub). This is especially helpful in clarifying how the “instantaneous” growth rates are calculated.

Reviewer #3 (Remarks to the Author):

Summary

In this work, the authors examine possible outcomes of competition among six species native to the respiratory tract. They find that depending on which species is initially dominant in the community, four different species may drive the others to extinction, producing four possible stable steady states. They then investigate to what extent this outcome is predictable from pairwise interactions. They find that pairwise interactions between four of the six species are highly bistable, with the outcome depending on which species is initially dominant. They find that this bistability is not caused by exclusively mutually antagonistic interactions between the pairs, since their metabolic byproducts do not universally suppress the other species. Instead, they identify density-dependent growth in most of the species, and conduct theoretical modeling to demonstrate that this can indeed drive bistability in a system. Theoretically and experimentally, using supplementation with the metabolite glutamate, they additionally show that weakening this density-dependent growth effect can lead to the collapse of such bistability.

General comments

This work is an elegant exploration of how density-dependent growth is sufficient to generate bistability in a system when competitive interactions are absent. This advances our understanding of the drivers of ecological stability. Modeling is used to recapitulate the qualitative experimental outcomes, including the effects of perturbations, which lends weight to the mechanisms suggested. The work also uses species relevant to a potential site of infection in humans. The link between the work and clinically relevant scenarios is somewhat murky but this to me is a minor point as the work itself is primarily a fundamental study addressing the mechanisms underlying the generation of metastability in microbial communities.

Specific Comments

- The work is rigorously carried out and well written, and it is a pleasure to read. I like how modeling was used to develop the basic concepts and to provide an intuitive interpretation of the qualitative aspects of the experimental data. Aspects can be further clarified. For instance, the work led by showing the generation of multiple states (Figure 1) emerging from the same initial configuration. The authors provide a brief description of the stochastic simulations that seem to capture the generation of multiple steady states. However, I feel this part should be described in more detail. For instance, did they only assume the Allee effect? Since they showed the importance of pairwise positive interactions, it would be good to show the impact of such cooperative interactions.
- Related to point 1, Figure 1 shows 6 replicates of the initial composition, which led to roughly 3 divergent outcomes. It would be nice to show more replicates, which may allow them to deduce with higher confidence for the relative frequency of the different outcomes. If so, I wonder whether the stochastic model can predict the relative frequency of the different outcomes.
- The data in extended data 8 could be alluded to earlier in the text; I was hoping when they were first introduced that the authors would comment on the Cn-Se mixture in 1c that is not represented in the stable states in 1d and whether it was transient or and unstable steady

state or something else.

- Given the focus on the importance of cooperative interactions, the authors should note a recent relevant paper (Wu et al, Nat Comm 2019: <https://doi.org/10.1038/s41467-018-08188-5>)
- In lines 210–214, the authors mention differing effects of the strength of the Allee effect. It might be helpful for the readers if this was shown graphically in 3d or the supplement.
- More analysis of Allee effect modulation is conducted in Figure 5ab. Some additional labels mentioning the strength of the Allee effect for N1 and N2 in the different scenarios would help readers get faster intuitive understanding of the different possibilities.
- Some more discussion on the clinical relevance of the findings (i.e. discussion of potential perturbations that might allow for the dominance of potential pathogens Sa/Cn) may be worth doing in the disc

POINT-BY-POINT RESPONSE TO THE REVIEWERS' COMMENTS

William Lopes, Daniel R. Amor and Jeff Gore

We would like to thank Jamila Roland-Chandler, Wenying Shou and the anonymous Reviewers for their valuable feedback. We believe that the manuscript has considerably improved after this revision. First, we have made changes in the introduction and discussion sections to enhance clarity and depth. Additionally, we have included new references to strengthen our literature review. We have also made modifications to figures 3 and 5 to improve their visual communication and ensure alignment with the text. Furthermore, we have included two new supplementary figures (Supplementary Figs. 2 and 6) to provide additional supporting information. In response to a suggestion made by Reviewer #1, we have reviewed the terminology used to describe within-species cooperative growth, aiming to avoid misunderstandings regarding other uses the word "cooperation". Below we reproduce the entire text from the Reviewers in plain gray text and include our response in bold. Additionally, new text included in the manuscript is highlighted in yellow.

Reviewer #1 (Remarks to the Author):

This review is written by Jamila Roland-Chandler under the supervision of Wenying Shou.

Lopes et al. investigated the drivers behind multistability in the nasal microbiome using experiments and modelling. The authors identified 6-species microbial communities had multiple single species steady states, which were driven by bistability in two-species coculture. Self-facilitation, aka Allee effects, rather than

mutual antagonism, drove bistability. Extending the Lotka-Volterra model to include Allee effects captured the observed dynamics in two-species coculture and predicted that weakening Allee effects would cause some loss of multistability and increased coexistence, which was validated experimentally. Therefore, Allee effects are important drivers of complex microbial community dynamics.

Lopes et al. explained a complex phenomenon in microbial community dynamics through experiments and modelling, and identified a novel mechanism for the emergence of multistability. Despite the complexity of the subject, the paper told a clear story that was easy to follow. Therefore, this paper should be accepted with minor corrections, which are detailed below.

We thank Jamila Roland-Chandler and Wenying Shou for their appreciation of our work and for providing us with valuable feedback.

The Lotka-Volterra model with Allee effects should be included in the main text.

We thank the Reviewer for this suggestion. We have included the equation for the Lotka-Volterra model with the Allee effects in the main text of the revised version, as follows:

Line 219: “We used a simple theoretical model to gain insight into how the Allee effect could shape outcomes of pairwise competitions between focal species. Specifically, we considered a modified version of the generalized Lotka-Volterra model that accounts for the Allee effect acting on some of the species (Methods). For species that are subject to the Allee effect, the dynamics is given by:

$$\frac{dX_i}{dt} = r_i X_i \left((X_i - a_i)(1 - X_i) - \alpha_{ij} X_j \right),$$

where $i \neq j$, X_i stands for abundance of species i (normalized to its carrying capacity), r_i stands for its *per capita* growth rate, α_{ij} captures the strength of the interaction from species j towards species i (with $i \neq j$) and a_i captures the strength of the Allee effect. Note that a strong or weak Allee effect is captured by a positive or negative value of a_i , respectively. For those species free of the Allee effect we consider dynamics given by the classical version of the Lotka-Volterra model, i.e. they are subject to logistic growth.”

The introduction overpromises what multistable dynamics are captured in this paper. In the introduction, Lopes et al. describe how human microbiomes can exhibit multistability. For example, gut microbiomes can switch from a high diversity to low diversity state after application of antibiotics (line 36, reference 8). However, this paper does not explain that sort of multistability, as the alternative stable states described in this paper are single or two species. A more appropriate example of multistability in the introduction should be used (e.g. reference 9 discussed bistability in the gut microbiome), or the discussion should include how the dynamics capture by this work could drive high and low diversity alternative steady states.

We acknowledge that the introduction would benefit from an example that better describes the existence of alternative stable states composed of strikingly distinct abundances of community members. We have changed the introduction as follows:

Line 35: (...) “Microbial communities can exhibit multistability, the potential for microbial abundances to reach alternative stable states under the same environmental conditions⁵. For instance, the human microbiome exhibits signatures of multistability⁶,

such as a bimodal distribution of *Prevotella* abundances in the gut, with one mode corresponding to the bacteria dominating the community, while in the other mode it is present in a low-abundance state⁷. In turn, the presence of multistability can enable short-term, temporary perturbations to induce sharp and lasting transitions in community composition and function⁸⁻¹². Persistent shifts in microbiome composition can significantly impact human health, potentially leading to unhealthy states that may prove difficult to reverse even with antibiotic treatment¹³. Both of these characteristics—dramatic shifts in species composition combined with difficulty in reversing the shift—are considered hallmarks of multistability¹⁴. Despite the fundamental importance of such alternative stable states, the drivers of multistability in microbial communities remain relatively unknown.”

Additionally, following the Reviewer’s feedback, we have decided to also change the discussion as follows:

Line 355: “The alternative stable states observed in this study consistently exhibit low biodiversity, constraining comparative analysis with typical states of the human microbiota. However, the Allee effect may nonetheless play a crucial role in driving alternative stable states in complex communities.”

Using the term “cooperation” to describe intra-species cooperation/self-facilitation/Allee effects is confusing. Cooperation in the literature is used to describe intra- or inter-species cooperation, but Lopes et al. mainly use it to describe intra-species cooperation. Although they do specify this, I think this terminology is confusing and should be changed to self-facilitation.

We appreciate this valuable suggestion. We acknowledge that the term “cooperation” might be overly broad and may not precisely align with the context of our study. In response to the Reviewer’s suggestion, the revised manuscript emphasizes our reference to within-species “cooperative growth” and minimizes the use of “cooperation” in any other context. We propose to keep the term “cooperative growth” in our work because of its frequent use in studies that address the Allee effect, while we also clarify its equivalence to “self-facilitation” through the manuscript.

Figure corrections

Fig. 1c and 5f: It is not clear which communities have reached steady state composition and which haven’t. I think there should be plots of community composition over each cycle, like fig. 2c, to make this clearer.

For Figs. 1c and 5f, we only measured final abundances due to technical constraints. Both experiments involved multiple communities, each with distinct initial fractions for every species, and comprised six replicates for each condition. Species abundances were assessed using CFU counting after plating on five different selective and differential media. Given the diverse conditions, we chose to focus on final abundances after 5 cycles, based on our lab experience that this approach, along with a 100-fold dilution, is usually sufficient to capture steady states. However, it is important to note that performing another growth-dilution cycle in Fig. 1c would potentially impact the abundance of species in replicates 5 and 6, likely leading to the exclusion of *Cn*. Nevertheless, it wouldn't affect the number of divergent outcomes which is the primary finding shown in Fig. 1c.

In the revised version of the manuscript, we have clarified this issue in the second paragraph of the results section:

Line 110: (...) "initializing C_n , M_c , S_a , or S_e at 95% of the community, we found that each of them could drive the five other species extinct, a result that was identical among the six community replicates (Fig. 1d). Furthermore, this observation suggests that communities starting at equal initial abundances might require more than 5 cycles to reach one of the four stable outcomes, as denoted by the presence of mutually exclusive species in the outcomes for communities 3 to 6 in Fig. 1c. Taken together, our findings uncover a surprisingly high degree of multistability in our six-species community, with four non-invadable and reproducible stable states."

Fig. 2c: In the C_n - S_a coculture, C_n increases from 1% to 80%, then declines to extinction. Why does this occur? I think this should be commented on.

We attribute the initial increase in C_n 's relative abundance (starting from a low abundance) to the distinct growth dynamics of the two species, which we elaborate on later in the manuscript. C_n grows logistically and relatively faster, while S_a is subject to a weak Allee effect. The presence of a weak Allee effect in S_a results in slower initial growth during the competition, providing an opportunity for C_n to grow in relative abundance before S_a reaches its maximum growth rate and carrying capacity.

In the revised version of the manuscript, we have addressed this observation as follows:

184: “We found that *Cn* is characterized by logistic growth (Supplementary Fig. 3), in which the *per capita* growth rate decreases monotonically with the population density. *Sa* and *Se* displayed a weak Allee effect, in which *per capita* growth rate is always positive but is reduced at low cell densities. Notably, distinct growth dynamics can affect the abundances of species over growth cycles in pairwise competitions. For example, in the *Cn-Sa* pair (Fig. 2c), the weak Allee effect in *Sa* results in slower initial growth at low abundance, allowing *Cn* to grow in relative abundance before *Sa* reaches its maximum growth rate and carrying capacity. Finally, we found that *Mc* is subject to a strong Allee effect, which is characterized by a negative growth rate at initial densities below a threshold ($\sim 10^5$ cell/mL).

Fig. 2d, 3d, 5d: The arrows used to indicate what steady states the system tends to are confusing. One could mistakenly think the arrows describe species interactions (like in fig. 3b). This representation should be changed.

To address the Reviewer's concerns, we have updated the arrow style used to visualize species interactions, as seen in Fig. 3b, thus allowing for clear differentiation from the arrows used in the network diagrams in Figs. 2d, 3d, and 5d. Additionally, we have explored alternative arrow styles and representations for the network diagrams; however, we believe that the current representation is still the simplest way to illustrate the pairwise outcomes, as it has been previously utilized by our group in similar contexts (Friedman et al., 2017; Lax and Gore, 2023). The updated version of Figure 3 follows:

Fig. 3| Most of the focal species exhibit intra- and inter-species facilitation, which could theoretically drive bistability through the Allee effect. **a**, Experimental scheme: Focal species were grown as monocultures for 24 h, and cells were filtered out from the culture. All focal species were grown in other species' spent media, as well as their own (see methods). **b**, Heat map of species' growth in spent media shows both inhibitory and facilitative interspecies interactions, as well as cooperative growth or "self-facilitation". Values are an average of at least three measurements. **c**, Growth rates for the focal species as a function of the initial population density (see methods) reveal a logistic growth pattern in Cn, whereas a positive but compromised growth rate at low densities indicates that both Sa and Se are subject to the weak Allee effect. Mc displays the strong Allee effect leading to population extinction below a survival threshold of 10^5 cell/mL. **d**, Exclusion and bistability outcomes as predicted by a Lotka-Volterra model in which either Se grows logistically (left) or exhibits a weak Allee effect (right). We model the growth of Cn as logistic in both cases. Dashed lines show the nullclines, boundaries between net positive and negative growth rates, for Se and Cn as a function of their normalized abundances. The area under each nullcline, matched in color,

indicates the region of positive *per capita* growth rate for the corresponding species. Arrows show the time trajectories of pairwise competitions starting from different initial abundances, empty dots show unstable steady states, and full dots show stable steady states.

Also, for figures such as 3d, adding colour-matched + (growth) or - (death) above and below the nullcline will help readers to grasp the message quickly.

We thank the Reviewer for this suggestion. It has been addressed as shown in Fig. 3 and further detailed in Fig. 5.

Fig. 5b and e: The shaded box indicating which pairwise cocultures exhibit theoretically-predicted dynamics is easy to miss. These results should be made more prominent. In addition, this information appears to be repeated in Fig 5e. Redundant information should be removed from the figure (e.g. add coculture names to fig. 5e like 5g).

We have modified the shaded boxes in Fig. 5b to make them more noticeable. Additionally, we implemented the suggestions provided by Reviewer #3 by adding labels to indicate the strength of the Allee effect for N1 and N2. Furthermore, we removed redundant information from the figure. The updated Fig. 5 follows:

Fig. 5| Theoretical prediction and experimental demonstration of a decrease in the number of alternative stable states when the strength of cooperative growth is reduced. Predicted and observed qualitative outcomes of pairwise competition, reshaped by reducing the strength of the Allee effect. **a**, The left panels show theoretical phase planes under the Allee effect (without glutamate supplementation), while right panels (**b**) show the same under reduced Allee effect (with glutamate supplementation). Nullclines, trajectories and steady states depicted by dashed lines, arrows and dots, respectively. The area under each nullcline, matched in color, indicates the region of positive *per capita* growth rate for the corresponding species, as shown in Fig. 3c. **c**, To understand how the modulation of cooperative growth impacts the alternative stable states, we co-cultured all pairs and the six-species community in media supplemented with glutamate. **d**, We found a strong reduction in the number of bistable pairs, as compared to Fig. 2d. The shaded box on the left indicates the species pairs for which we observed the predicted reshaping of competition outcomes. **e**, Under reduced Allee effect by glutamate supplementation, the six-species community displays two alternative stable states in which two species coexist (gray circles 1 and 2), as predicted from simple assembly rules based on pairwise competition outcomes²¹.

Reviewer #2 (Remarks to the Author):

Analysis of the stable states is a central question in the study of microbial communities. This article looks into the multistability of synthetic microbial communities assembled from six species found in the respiratory tract. Bistability is commonly assumed to be driven by a mutually inhibitory interaction between two species (-/- interaction). This study, however, reports the presence of bistability in cooperating species (+/+) and in predator-prey type interaction (-/+), which is an interesting observation. The authors argue that cooperative growth can be a driving mechanism of bistability. They test and report the presence of cooperative growth among these species. Next, they use glutamate supplementation to tune down the level of cooperative growth which results in a reduction in the number of alternative stable states.

This article is generally well-written, clear, and easy-to-follow. Evidence is offered with well-designed experiments. We have the following minor comments that we think will be helpful for readers.

We thank the Reviewer for the kind comments regarding our work.

Minor comments:

1. The interaction values, α_{ij} , are all set to be positive which represent two competing species. However, the CFSM experiments in Fig 3b shows the presence of facilitation in many pairs. We would suggest additionally running the theoretical experiments under non-competing settings. In other words, setting the α_{ij} value in a way that its sign (- or +) matches Fig 3b.

In the revised manuscript, we explain that in the case of a weak Allee effect, bistability requires competitive interactions, whereas this is not the case with a strong Allee effect. Moreover, we have included "Supplementary Fig. 6" which analyzes the theoretical outcomes when the sign of interactions is chosen as suggested by the spent media experiments in Fig. 3b. The revised manuscript includes the following text:

Line 235: "While a strong Allee effect can generate bistability in pairwise outcomes regardless of the sign of interspecies interactions, the weak Allee effect can only drive such bistability under competitive interspecies interactions (Supplementary Fig. 6)."

The new Supplementary Figure 6 is included as follows:

Supplementary Fig. 6] The LV model with cooperative interspecies interactions fails to capture the experimentally observed bistability driven by the weak Allee effect.

Results from spent media experiments in Fig. 3b suggest that different pairs of species could exhibit facilitative interspecies interactions, which raises the question of how such facilitative interactions impact pairwise outcomes in the theoretical model. The panels show the nullclines, the stable (full dots) and unstable (empty dots) steady states, and representative trajectories (lines with arrows) in the phase space for different species pairs. The area under each nullcline, matched in color, indicates the region of positive *per capita* growth rate for the corresponding species. Here we chose the sign of interspecies interactions according to the outcomes of spent media experiments (Fig. 3b). For example, $a_{ij} < 0$ for $i=Cn$ and $j=Se$, since we observed that the spent media of *Se* facilitates the growth of *Cn*. **a**, Phase space for *Cn* and *Se*, with *Se* exhibiting either a weak Allee effect (left) or logistic growth (right). The cooperative interspecies interactions play a stronger role in the outcome than the weak Allee effect, as evidenced by the relatively similar shape of the nullclines in the two panels. In the case of logistic growth, however, trajectories are

shorter and the species pair reaches the steady state faster. **b**, Phase space for S_e and S_c for both species under a weak Allee effect (left) or logistic growth (right). As in **a**, the system has a unique stable state in both cases—note that the interspecies cooperation enables the stable abundances to exceed the normalized carrying capacity. **c and d**, Under the strong Allee effect acting on M_c , the Allee threshold determines the competition outcome in addition to the interaction parameters. For a reduced Allee effect acting on M_c , M_c excludes C_n , while the system is bistable when M_c exhibits the strong Allee effect or competes against S_e or S_a . Note that, due to the change in the interspecies interaction sign, the parabolic nullcline enables regions in which M_c can grow no matter how high the abundance of the other species is. Overall, the LV model does not predict bistability driven by a species' weak Allee effect if the interspecies interactions towards this species are cooperative. Our choice of using competitive interactions in the main text was based on: *i*) assessing interspecies interactions through experimental pairwise outcomes suggests that competition is present in co-cultures, *ii*) the LV model can lead to unrealistic, unbounded growth in the presence of cooperative interspecies interactions, and *iii*) spent media experiments are more suited to assessing intra-species interactions than inter-species interactions, since the spent media are generated by cells growing in the presence of the relevant interacting cell type (i.e. cells from the same species) in the case of intra-species interactions, but this is not the case in the case when inferring interspecies interactions—in the latter case, one monoculture produces the spent media that will be tested in a different species monoculture, potentially missing metabolites that are only produced in the presence of the other species¹.

2. The authors have observed three cases of strong/weak Allee effects and one case of logistic growth among the four main species that primarily determined the stable states. We think it would be helpful if they can speculate whether these proportions are commonly expected. Under what conditions?

In alignment with the Reviewer's observation of the relevance of these findings, we were surprised by the proportion of species subject to the Allee effects in our system. For instance, microbial communities might exhibit the Allee effects more prominently under certain conditions, where extracellular degradation of complex nutrients is required for growth. In our study, we conducted experiments using Brain Heart Infusion (BHI) media at a 10% concentration, which is generally considered a chemically rich medium supporting microbial growth, although its specific nutrient composition is unknown.

Previously, our group uncovered distinct mechanisms that drive cooperative growth in microbial populations. We demonstrated that the yeast *Saccharomyces cerevisiae* is subject to a weak Allee effect in media containing sucrose, in which external factors such as daily dilution rates can significantly influence the strength of this effect, potentially transitioning it from weak to strong and vice versa (Gore et al., 2009). Additionally, we demonstrated that deactivating antibiotics, such as ampicillin, can promote cooperative growth dynamics (Yurtsev et al., 2013). These laboratory-derived examples suggest that the mechanisms underlying the Allee effect might be prevalent enough to occur commonly across natural microbial communities. Nonetheless, accurately predicting its extent remains a challenging task.

3. In their survey the authors found an important role for glutamate. Is this a general expectation? Is the concentration of glutamate particularly low or high in the respiratory tract? Alternatively, is BHI selectively low in glutamate? It would be nice to link back the observation to either the ecology of the respiratory tract or the particular laboratory conditions used for these investigations.

In our study, we indeed uncovered a significant role for glutamate, shedding light on its importance in the dynamics and assembly of microbial communities. Notably, glutamate is the most abundant intracellular metabolite in *Escherichia coli* and serves a key function in regulating both population density and growth rate (Mukherjee, et al. 2023). Additionally, studies have demonstrated that glutamate can be metabolized and produced by some commensal bacteria, thereby potentially impacting numerous metabolic processes in mammalian hosts (Mazzoli and Pessione, 2016). In this context, it is expected that significant roles for glutamate will continue to be discovered.

In the respiratory tract, including nasal secretions, the concentration of glutamate is particularly high in comparison to other amino acids (Hilding et al., 1973), ranging from 39.8 to 103.4 μ M (Krismer et al., 2014).

We could not find quantitative data about the glutamate content in BHI, likely because it is neither defined nor uniform in composition. Nevertheless, the presence of beef extract and infusion components in BHI certainly contributes to the overall glutamate content.

We have added the following text to better address the importance of glutamate in the ecology of the respiratory tract:

Line 367: "Given that the concentration of glutamate in nasal secretions is high in comparison to other amino acids^{42,43}, it is plausible to assume that glutamate may influence the growth dynamics and long-term survival of some microbes colonizing the nasopharynx."

4. We would suggest including the raw data of the monocultures and cocultures used for the figures in the manuscript (either visually as supplementary figures or as a link to an openly available repository such as GitHub). This is especially helpful in clarifying how the "instantaneous" growth rates are calculated.

We have uploaded all raw data to our GitHub account

Reviewer #3 (Remarks to the Author):

In this work, the authors examine possible outcomes of competition among six species native to the respiratory tract. They find that depending on which species is

initially dominant in the community, four different species may drive the others to extinction, producing four possible stable steady states. They then investigate to what extent this outcome is predictable from pairwise interactions. They find that pairwise interactions between four of the six species are highly bistable, with the outcome depending on which species is initially dominant. They find that this bistability is not caused by exclusively mutually antagonistic interactions between the pairs, since their metabolic byproducts do not universally suppress the other species. Instead, they identify density-dependent growth in most of the species, and conduct theoretical modeling to demonstrate that this can indeed drive bistability in a system. Theoretically and experimentally, using supplementation with the metabolite glutamate, they additionally show that weakening this density-dependent growth effect can lead to the collapse of such bistability.

General comments

This work is an elegant exploration of how density-dependent growth is sufficient to generate bistability in a system when competitive interactions are absent. This advances our understanding of the drivers of ecological stability. Modeling is used to recapitulate the qualitative experimental outcomes, including the effects of perturbations, which lends weight to the mechanisms suggested. The work also uses species relevant to a potential site of infection in humans. The link between the work and clinically relevant scenarios is somewhat murky but this to me is a minor point as the work itself is primarily a fundamental study addressing the mechanisms underlying the generation of metastability in microbial communities.

We thank the Reviewer for appreciating our work. We have addressed the potential clinical relevance of the work in the responses below.

Specific Comments

- The work is rigorously carried out and well written, and it is a pleasure to read. I like how modeling was used to develop the basic concepts and to provide an intuitive interpretation of the qualitative aspects of the experimental data. Aspects can be further clarified. For instance, the work led by showing the generation of multiple states (Figure 1) emerging from the same initial configuration. The authors provide a brief description of the stochastic simulations that seem to capture the generation of multiple steady states. However, I feel this part should be described in more detail. For instance, did they only assume the Allee effect? Since they showed the importance of pairwise positive interactions, it would be good to show the impact of such cooperative interactions.

We appreciate the insightful suggestions and comments provided by the Reviewer. In addressing a question raised by Reviewer #2, we have clarified that we considered competitive interactions in our study. Given that the deterministic model fails to predict bistability driven by a weak Allee effect under cooperative interspecies interactions, this choice of interactions also fails to capture the experimental observations in the stochastic model. We have added a sentence in the Main Text, citing the new Supplementary Fig. 6 (as detailed in our response to Reviewer #2 above), to better explain this point in our revised manuscript. Additionally, the captions of Supplementary Figs. 10 and 11 now clearly state

that our stochastic simulations account for competitive interactions and detail which species are affected by either weak or strong Allee effects.

The changes in the mentioned figure captions are highlighted below:

Supplementary Fig. 10| Sensitivity to stochastic fluctuations can lead to divergent community outcomes from the same initial conditions. The panels display representative results from simulations of the stochastic model considering competitive interspecies interactions, as well as a weak Allee effect acting on S_a and S_e , and a strong Allee effect acting on M_c (Supplementary Text 1). (...)

Supplementary Fig. 11| Reducing the Allee effect in stochastic populations recapitulates the observed changes in the number of alternative stable states. The panels display representative results from simulations of the stochastic model, which account for competitive interspecies interactions. This scenario considers only M_c subject to a strong Allee effect, while S_a and S_e , as well as C_n , exhibit logistic growth (Supplementary Text 1). (...)

- Related to point 1, Figure 1 shows 6 replicates of the initial composition, which led to roughly 3 divergent outcomes. It would be nice to show more replicates, which may allow them to deduce with higher confidence for the relative frequency of the different outcomes. If so, I wonder whether the stochastic model can predict the relative frequency of the different outcomes.

We appreciate the Reviewer's comment. We acknowledge that it would be interesting to deduce with confidence relative frequencies of these outcomes. Nevertheless, it is important to take into account that the lack of qualitative reproducibility is inherent in such experiments. When mixing species at equal abundances, even small variations in initial inoculum density may lead to significant divergence across the replicates, possibly influenced by the system's separating basins of attraction, where the dynamics pose a challenge in accurately predicting probabilities with a high degree of confidence. Thus, the relative frequencies of the different outcomes are expected to be different

among different biological replicates. Considering that demonstrating the Allee effect as a mechanism driving multistability remains the main aim of our work, we believe that exploring the relative frequency of the different outcomes (at equal initial densities for the 6-species) may be better suited for another context.

- The data in Supplementary 8 could be alluded to earlier in the text; I was hoping when they were first introduced that the authors would comment on the Cn-Se mixture in unstable steady state or something else.

We thank the Reviewer for this suggestion. After thorough consideration, we propose to keep the Supplementary Fig. 8 (currently Supplementary Fig. 10) later in the text. We believe that readers who may not be familiar with the modeling of the Allee effect will find it easier to begin with a description of the deterministic model. Otherwise, readers would be presented with a more complex model at an earlier stage, potentially making it more challenging to understand the main drivers underlying community dynamics. Nevertheless, our response to the figure corrections proposed by Reviewer #1 addresses the stability of the Cn-Se mixture, considering the possibility that it might be an unstable outcome. The specific correction is included below:

Line 110: (...) "initializing C_n , M_c , S_a , or S_e at 95% of the community, we found that each of them could drive the five other species extinct, a result that was identical among the six community replicates (Fig. 1d). Furthermore, this observation suggests that communities starting at equal initial abundances might require more than 5 cycles to reach one of the four stable outcomes, as denoted by the presence of mutually exclusive species in the outcomes for communities 3 to 6 in Fig. 1c. Taken together,

our findings uncover a surprisingly high degree of multistability in our six-species community, with four non-invadable and reproducible stable states.”

- Given the focus on the importance of cooperative interactions, the authors should note a recent relevant paper (Wu et al, Nat Comm 2019: <https://doi.org/10.1038/s41467-018-08188-5>)

We thank the Reviewer for pointing us to this relevant reference. In the revised version, we have cited it (Ref. 35) in the discussion section as follows:

Line 343: “Coarse-grained rules have also proven effective in predicting outcomes such as collapse, coexistence and productivity, across models and experiments for mutualistic communities³⁵. Guided by coculture outcomes (Fig. 2c), we used competitive interspecies interactions to theoretically analyze the impact of cooperative growth in bistable pairs (with an extension to mutualistic pairs provided in Supplementary Fig. 6). Future work could focus on addressing the impact of cooperative growth in experimental, mutualistic communities.”

- In lines 210-214, the authors mention differing effects of the strength of the Allee effect. It might be helpful for the readers if this was shown graphically in 3d or the supplement.

We thank the reviewer for this suggestion, as we also believe that including a cartoon about the Allee effects will improve the overall comprehension of the work. We have added a new Supplementary Figure that illustrates the different Allee effects:

Supplementary Fig. 2| Cartoon illustrating different within-species population dynamics. a, Biological populations following logistic growth exhibit intraspecific competition, growing best at lower densities and experiencing a linearly decreasing *per capita* growth rate with population density. Conversely, some populations exhibit slower growth at low densities, growing best at intermediate densities due to intraspecific cooperation, known as the Allee effect. The Allee effect can vary in strength, being weak **(b)**, and therefore resulting in a reduced but positive growth rate at low density, or strong **(c)**, leading to a threshold abundance for survival below which the growth rate becomes negative, potentially leading the population to extinction.

- More analysis of Allee effect modulation is conducted in Figure 5ab. Some additional labels mentioning the strength of the Allee effect for N1 and N2 in the different scenarios would help readers get faster intuitive understanding of the different possibilities.

We have updated Fig. 5 by including labels indicating the strength of the Allee effect for N1 and N2. These improvements can be seen as part of our response to Reviewer #1's comments regarding Figure 5.

- Some more discussion on the clinical relevance of the findings (i.e. discussion of potential perturbations that might allow for the dominance of potential pathogens Sa/Cn) may be worth doing in the disc

We have included the following text in the discussion to highlight the biological relevance:

Line 372: "The community considered here provides a simplified representation of communities that play a critical role in human health. Of notable clinical relevance, approximately 30% of healthy people asymptotically carry a strain of *Sa* in the anterior nares⁴⁵, a significant observation, since the vast majority of cases of *Sa* bloodstream infection appear to be endogenous, involving the same strain as the nasal colonizer^{46,47}. Moreover, several studies have shown that during infections such as pneumonia, there is a reduced diversity in the respiratory microbiome often marked by increased abundance of *Mc*. Following an infection, individuals may establish a long-term, healthy microbiome with a composition shift towards the dominance of *Mc*^{48,49}."

REVIEWERS' COMMENTS

Reviewer #2 (Remarks to the Author):

In my opinion, the authors have adequately addressed the concerns raised by all referees.

Reviewer #3 (Remarks to the Author):

The authors have fully addressed my raised issues and have further strengthened the paper, which was already strong.

Reviewer #4 (Remarks to the Author):

The authors have adequately addressed all the concerns raised by the referees. We have no additional comments.